# Two Failures of Self-Consistency in the Multi-Step Reasoning of LLMs

**Angelica Chen**                                                    *ac5968@nyu.edu*
*Center for Data Science, New York University*

**Jason Phang**                                                      *zp489@nyu.edu*
*Center for Data Science, New York University*

**Alicia Parrish**                                                   *avp295@nyu.edu*
*Department of Linguistics, New York University; Google*

**Vishakh Padmakumar**                                              *vp1271@nyu.edu*
*Center for Data Science, New York University*

**Chen Zhao**                                                        *cz1285@nyu.edu*
*NYU Shanghai; Center for Data Science, New York University*

**Samuel R. Bowman**                                                *sb6065@nyu.edu*
*Center for Data Science, New York University; Anthropic*

**Kyunghyun Cho**                                                    *kc119@nyu.edu*
*Center for Data Science, New York University*

**Reviewed on OpenReview:** *https://openreview.net/forum?id=5nBqY1y96B*

## Abstract

Large language models (LLMs) have achieved widespread success on a variety of in-context few-shot tasks, but this success is typically evaluated via correctness rather than consistency. We argue that self-consistency is an important criteria for valid multi-step reasoning in tasks where the solution is composed of the answers to multiple sub-steps. We propose two types of self-consistency that are particularly important for multi-step reasoning – hypothetical consistency (a model's ability to predict what its output would be in a hypothetical other context) and compositional consistency (consistency of a model's final outputs when intermediate sub-steps are replaced with the model's outputs for those steps). We demonstrate that multiple variants of the GPT-3/-4 models exhibit poor consistency rates across both types of consistency on a variety of tasks.

## 1 Introduction

An important property of logically valid machine learning systems is *self-consistency – i.e.*, the requirement that no two statements given by the system are contradictory. Pre-trained large language models (LLMs), despite demonstrating impressive few-shot accuracy on a variety of multi-step reasoning tasks, often give inconsistent responses to questions (Mitchell et al., 2022; Kassner et al., 2021) and factual knowledge-seeking prompts (Elazar et al., 2021). Without self-consistency, it is difficult to consider LLMs reliable or trustworthy systems. Elazar et al. (2021) defines self-consistency as the invariance of an LLM's responses across different types of semantics-preserving *prompt transformations*. In this work, we seek to introduce and explore LLM self-consistency over two new types of transformations (shown in Figure 1) that we argue are important for valid multi-step reasoning.

**Hypothetical Transformations**  A *hypothetical transformation* is an indirect phrasing of a prompt that queries the model for what its response would hypothetically be in some other context, such as "what would your response to `<prompt>` be?" or "what would the next 5 words in your completion of `<prompt>` be?" Consistency over hypothetical transformations implies that an LLM has some stored knowledge or computational path for determining what its response would be to some prompt $p$ without explicitly being prompted with exactly $p$ itself. This can be useful for prompts involving multi-step reasoning, where the LLM must have knowledge of its responses to the earlier steps in order to compute its responses to downstream steps. Like in Figure 1, given the prompt "What is the runtime of the best movie of 2022" the LLM must either have stored or computed its response to "what is the best movie of 2022?" in order to answer the full prompt.

**Compositional Transformations**  For a prompt that involves multiple interdependent steps of reasoning, a *compositional transformation* consists of replacing some intermediate step with the model's output to the previous step. In the previous example, if the LLM outputs the response "*Everything Everywhere All At Once*" to the prompt "What is the best movie

Figure 1: An overview of the two types of self-consistency failures we identify in LLMs.

of 2022?," then the prompt "What is the runtime of *Everything Everywhere All At Once*?" is a compositional transformation of "What is the runtime of the best movie of 2022?" (See Figure 1.) Consistency over compositional transformations is also important for logically valid multi-step reasoning when the LLM must give a direct response – without it, the LLM may give contradictory direct responses to different multi-step prompts that are in fact querying for the same thing.

In this work, we investigate the degree to which LLMs are self-consistent on these prompt transformations across a variety of tasks. We formalize our definitions of hypothetical and compositional consistency (Section 2) and show empirically that a wide range of pre-trained language models demonstrate low consistency rates on both hypothetical transformations (Section 3) and compositional transformations (Section 4).

## 2  Formalizing Consistency

To make more precise our definitions of consistency and the semantics-preserving transformations that they entail, we first formalize our definitions prior to conducting our empirical evaluations.

### 2.1  Preliminaries

Let vocabulary $\mathcal{V}$ be a finite set of tokens, $\mathcal{V}^*$ be the set of all possible finite-length sequences formed by concatenating zero or more tokens from $\mathcal{V}$, and $p_\theta : \mathcal{V} \to \{0, 1\}$ be an auto-regressive language model that defines a probability distribution over tokens $v \in \mathcal{V}$. $p_\theta$ can be used to generate a sequence via greedy decoding as follows:

$$\tilde{y}_t = \arg\max_{v \in \mathcal{V}} \log p_\theta(y_t = v \,|\, c; \tilde{y}_{<t}) \tag{1}$$

given some context sequence $c \in \mathcal{V}^*$, until some time step $T$ for which $\tilde{y}_T = $ `[EOS]`, the end-of-sequence token. For ease of notation, we denote the greedy decoding output of a model $p_\theta$ as

$$g_{p_\theta}(c) = (\,\arg\max_{v \in \mathcal{V}} p_\theta(y = v|c), \cdots, \tag{2}$$

$$\arg\max_{v \in \mathcal{V}} p_\theta(y = v|c; \tilde{y}_{<T})).$$

We also define an operator $\sim$ that indicates when two strings are *semantically equivalent*. Although the precise definition of semantic equivalence will vary across different tasks, we use it to loosely refer to pairs of strings that can be used interchangeably (give or take syntactic adjustments) without changing the meaning of the overall utterance. Lastly, the $\sim$ operator is also reflexive, symmetric, and transitive.

## 2.2 Composing prompts

Reasoning with language often also involves composing prompts – for instance, we might ask "what is the answer to $2 \times 3 + 4$?", which can be seen as the composition of a *prompt template* "what is the answer to $\_ + 4$?" with the prompt "$2 \times 3$", where the "$\_$" symbol in the former string is substituted with the latter string. This corresponds to a multi-step task where the model might first answer the prompt "$2 \times 3$" (yielding $g_{p_\theta}("2 \times 3")$), substitute $g_{p_\theta}("2 \times 3")$ into the template (yielding the composed prompt "what is the answer to $g_{p_\theta}("2 \times 3") + 4$?," where the $g_{p_\theta}("2 \times 3")$ is replaced with the actual output string), and then answer the filled-in template.

To denote such prompt templates, we define $\mathcal{P}'$, the set of prompts $p \in \mathcal{V}^*$ that contain exactly one "$\_$" symbol. Additionally, the function $f(p', p) : \mathcal{P}' \times \mathcal{V}^* \to \mathcal{V}^*$ denotes substitution of $p$ for the "$\_$" symbol in $p'$.[1]

$f$ also has some useful properties that we will use in our later definitions:

- We can trivially represent any prompt $p \in \mathcal{V}^*$ as the substitution of itself into the *identity prompt template* "$\_$" by writing $p = f("\_", p)$.

- $p \sim q$ if and only if $f(p', p) \sim f(p', q)$ for all $p' \in \mathcal{P}'$.

## 2.3 Definitions

We start out by restating the general definition of self-consistency, as it has been commonly defined in past literature (Elazar et al., 2021; Jang et al., 2022).

**Definition 2.1** (Self-consistency). $p_\theta$ is *self-consistent* if $p \sim q \to g_{p_\theta}(p) \sim g_{p_\theta}(q)$ for all $p, q \in \mathcal{V}^*$.

In other words, a self-consistent model gives semantically-equivalent responses to semantically equivalent prompts. These semantically equivalent pairs of prompts ($p, q$ in Definition 2.1) can take many forms, including hypothetical and compositional transformations.

**Definition 2.2** (Hypothetical Transformation). Let $\mathcal{P}'_I$ denote the set of *hypothetical transformation prompt templates*, which are prompt templates $p' \in \mathcal{P}'$ such that $f(p', p) \sim f(\_, p) \, \forall p \in \mathcal{V}^*$. Then the set of *hypothetical transformations* of prompt $p$ can be denoted as $\mathcal{P}_I(p) := \{f(p', p) \, | \, p' \in \mathcal{P}'_I\}$.

Since $f(\_, p) \sim p$, a model that is self-consistent must yield $g_{p_\theta}(f(p', p)) \sim g_{p_\theta}(p)$ for all $p' \in \mathcal{P}'_I$.

Although we defined hypothetical transformations with respect to all prompts $p \in \mathcal{V}^*$, our definition of compositional transformations must be more restricted, since we care only to apply compositional transformations to prompts that implicitly encode a compositional task. That is, we are concerned only with prompts that are already compositions – *i.e.*, prompts of the form $f(p', p)$. Furthermore, given some target model $p^*$ that represents the ground truth or gold distribution, the response to a compositional prompt (as generated by $p^*$) is semantically equivalent to the prompt itself. That is, $f(p', p) \sim g_{p^*}(f(p', p))$. For example, if $p = "2"$ and $p' = "4 + \_"$, then $f(p', p) = "4 + 2"$ and $g_{p^*}(f(p', p)) = g_{p^*}("4 + 2") = 6 \sim "4 + 2" = f(p', p)$, so $f(p', p)$ is compositional.

**Definition 2.3** (Compositional prompt). We define the set of *compositional prompts* as $\mathcal{P}_{\mathrm{Comp}} := \{f(p', p) \, | \, f(p', p) \sim g_{p^*}(f(p', p)), p \in \mathcal{V}^*, p' \in \mathcal{P}'\}$ given gold distribution $p^*$.

**Definition 2.4** (Compositional transformation). For prompt compositions $p \in \mathcal{P}_{\mathrm{Comp}}$ and $f(p', p) \in \mathcal{P}_{\mathrm{Comp}}$ both representing compositional tasks, the *compositional transformation* with respect to model $p_\theta$ is $f(p', g_{p_\theta}(p))$.

---

[1]In practice, substituting $p$ into $p'$ may require minor syntactic adjustments for linguistic acceptability, but we omit these in our notation since the semantics remain the same.

Given the above two types of prompt transformations, we can define narrower types of LLM self-consistency.

**Definition 2.5** (Hypothetical consistency). A model $p_\theta$ is *hypothetically consistent* if $g_{p_\theta}(p) \sim g_{p_\theta}(f(p', p))$ for any prompt $p \in \mathcal{V}^*$ and hypothetical transformation prompt template $p' \in \mathcal{P}'_I$.

**Claim 2.6.** If $p_\theta$ is self-consistent, then $p_\theta$ is also hypothetically consistent.

*Proof.* Consider prompt $p \in \mathcal{V}^*$ and hypothetical transformation prompt template $p \in \mathcal{P}'_I$. Since $f(p', p) \sim f(\_, p)$ (by Definition 2.2) and $f(\_, p) \sim p$, then $f(p', p) \sim p$ by the transitive property of $\sim$. Then by Definition 2.1, it follows that $g_{p_\theta}(f(p', p)) \sim g_{p_\theta}(p)$. $\qquad\square$

**Definition 2.7** (Consistency over compositional transformations). A model $p_\theta$ is *compositionally consistent* when, for all pairs of compositional prompts $p \in \mathcal{P}_{\text{Comp}}$ and $f(p', p) \in \mathcal{P}_{\text{Comp}}$,

    1. $p \sim g_{p_\theta}(p)$ and $g_{p_\theta}(f(p', p)) \sim g_{p_\theta}(f(p', g_{p_\theta}(p)))$

and is *compositionally inconsistent* when:

    1. $p \nsim g_{p_\theta}(p)$ and $g_{p_\theta}(f(p', p)) \sim g_{p_\theta}(f(p', g_{p_\theta}(p)))$

    2. $p \sim g_{p_\theta}(p)$ and $g_{p_\theta}(f(p', p)) \nsim g_{p_\theta}(f(p', g_{p_\theta}(p)))$

One more case exists that we do not count into our measures of either compositional consistency or inconsistency. If $p \nsim g_{p_\theta}(p)$ and $g_{p_\theta}(f(p', p)) \nsim g_{p_\theta}(f(p', g_{p_\theta}(p)))$, then we cannot say this is compositionally consistent because the model's output for $p$ does not necessarily relate to or imply its output to $f(p', p)$. However, we also cannot necessarily say that it is compositionally inconsistent to give nonequivalent responses to $f(p', p)$ and $f(p', g_{p_\theta}(p))$ if the model's responses to $p$ and $g_{p_\theta}(p)$ are also nonequivalent.

# 3 Evaluating Consistency on Hypothetical Transformations

In our above definitions, hypothetical consistency is characterized as a binary – either a model is hypothetically consistent across *all* pairs $(p, p') \in \mathcal{V}^* \times \mathcal{P}'_I$ or it is hypothetically inconsistent. But in practice, models are only hypothetically consistent in some cases, and it is likely impossible to achieve hypothetical consistency across all $(p, p') \in \mathcal{V}^* \times \mathcal{P}'_I$. Instead, we explore the *degree* to which LLM outputs are invariant to hypothetical transformations of the prompt. This is measured as the **hypothetical consistency rate**, which is the proportion of pairs $(p, p') \in \mathcal{V}^* \times \mathcal{P}'_I$ for which a model $p_\theta$ exhibits the property $g_{p_\theta}(p) \sim g_{p_\theta}(f(p', p))$. To measure hypothetical consistency rate, we devise a set of four hypothetical transformation prompt templates that we use to transform randomly sampled prompts (sourced from Wikipedia and DailyDialog) into hypothetical prompts, as shown in Table 1. We average the hypothetical consistency rate over these prompts to mitigate the model's sensitivity to prompt wording (Elazar et al., 2021).

Evaluating hypothetical consistency requires checking whether $g_{p_\theta}(p) \sim g_{p_\theta}(f(p', p))$. However, checking this semantic equivalence is non-trivial – the same idea can be expressed in a number of different but synonymous ways. Rather than attempt to devise an automatic method for evaluating semantic equivalence, we instead use a multiple-choice set-up. One answer choice is the continuation of the initial prompt (denoted by "`<prompt>`") sourced from a text dataset, one choice is the model's own greedily decoded completion for `<prompt>`, and the three remaining choices are the other models' completions for `<prompt>`. As discussed before, a model that is hypothetically consistent can, in a sense, predict its own completion. Thus, the model should be more likely to generate the answer choice that corresponds to its own completion than to the other answer choices. These templates are designed both to query the model on what its completion would hypothetically be for a given prompt and to evaluate whether the model can distinguish its own completions from those of other models.

As an example, suppose the original prompt sourced from Wikipedia is "This quilt begun in 1856 when she was seventeen includes the autographs on top of the blocks of many known celebrities and politicians of the day. Other". Suppose that the first three words of the completions generated by OpenAI GPT-3 models `ada-001`, `babbage-001`, `curie-001`, and `davinci-003` are "notable quilt authors," "famous quilts include,"

"signatures include abolitionists," and "notable figures whose," respectively. Additionally, the next three words of the Wikipedia article are "figures represented on." Then a hypothetical transformation prompt that uses the first template in Table 1 might look like:

> I predict that the next 3 words after "This quilt begun in 1856 when she was seventeen includes the autographs on top of the blocks of many known celebrities and politicians of the day. Other" would be
>
> A) famous quilts include
> B) figures represented on
> C) notable quilt authors
> D) signatures include abolitionists
> E) notable figures whose
>
> Answer:

If the model being evaluated is `davinci-003`, then the correct answer would be E. In this context, $g_{\texttt{davinci-003}}$("I predict that... Other" would be") = "E" $\sim$ "notable figures whose" = $g_{\texttt{davinci-003}}$("This quilt begun... Other"), which satisfies Definition 2.5. As mentioned in Section 3.1, all hypothetical prompts are few-shot (where the provided labels are the answer choices that correspond to the evaluated model's completion), and only case-insensitive exact-match answers (*i.e.* "a/A/b/B/c/C/d/D") are accepted as correct.

Although this experimental set-up eases the burden of checking the semantic equivalence of $g_{p_\theta}(p) \sim g_{p_\theta}(f(p', p))$, it is likely still a lower bound on the true hypothetical consistency, since it does not account for other model outputs that may be semantically equivalent to $g_{p_\theta}(p)$. In the example above, had `davinci-003` generated "notable people whose" instead of the letter "E," this response would have still been marked incorrect despite the semantic equivalence of "notable figures whose" and "notable people whose." Furthermore, the definition of semantic equivalence here depends implicitly on the prompt $p$. Nevertheless, we still find it useful to analyze these lower bounds – examinations of the failure modes of LLMs can aid in the future improvements of these models. We leave to future work the question of how far this lower bound is from the true hypothetical consistency rate.

Table 1: Prompt templates used to evaluate whether LLMs are consistent across hypothetical prompt transformations. `<prompt>` is a prompt sourced from a dataset (e.g. Wikipedia, DailyDialog), `<m>` is the number of words of its own completion that the model is asked to predict, and `<answer_choices>` are the multiple-choice answer choices that the model is given.

| **Hypothetical Transformation Prompt Templates** |
| --- |
| "I predict that the next `<m>` words after "`<prompt>`" would be `<answer_choices>`. Answer:" |
| "Given the prompt "`<prompt>`", my next `<m>` words would be `<answer_choices>`. Answer:" |
| "Given the context "`<prompt>`", my next `<m>` words would be `<answer_choices>`. Answer:" |
| "I predict that after the context "`<prompt>`" the next `<m>` words I would say are `<answer_choices>`. Answer:" |

We conduct our hypothetical consistency experiments with original prompts sourced from two language modeling tasks:

- **Wikipedia** Since language models are frequently pre-trained on Wikipedia archives, evaluating on a Wikipedia dataset can confound information memorized during pre-training with the skill being evaluated. To address this issue, we collect a sample of 400 English Wikipedia (Wikimedia Foundation) articles that were created on or after June 30, 2021, since the OpenAI documentation (Center) indicates that the latest pre-training data for the `ada-/babbage-/curie-/davinci-001` and `davinci-003` models contains documents dating up to June 2021. Each initial prompt is a randomly

selected segment of a Wikipedia article consisting of two full sentences followed by a random-length prefix of the next sentence.

- **DailyDialog**: DailyDialog (Li et al., 2017) is a manually labeled dataset of multi-turn conversations about daily life. We choose this dataset because it contains language that is more colloquial in style and less factual in content than the Wikipedia dataset. We randomly sample 400 examples from the training split and use the first conversational turn as the initial prompt.

We use only the prompts for which all five answer choices are distinct. We also vary the number of words $m$ in the original completion that the model is asked to distinguish from 1 to 6, since the difficulty may vary depending on the length of the original completion. We then compute the hypothetical consistency rate by calculating the proportion of the time that the model generated the letter of the answer choice corresponding to its own completion.

## 3.1 Experimental Setup

In all experiments, we evaluate four model sizes of the OpenAI GPT-3 model (Brown et al., 2020) – `ada-001`, `babbage-001`, `curie-001`, and `davinci-003` (in order of increasing capacity).[2] All experiments are run using greedily decoded completions obtained from the OpenAI API from Aug. 2022 to Jun. 2023. We use 0-shot initial prompts but evaluate hypothetical consistency prompts using $k$-shot prompts, where $k$ ranges from 1 to 10. Since the in-context performance of LLMs is known to vary depending on the selection of in-context examples (Liu et al., 2021; Rubin et al., 2022), we randomly select a different set of in-context examples for each prompt. We also randomize the order of answer choices for each multiple-choice question to mitigate sensitivity to answer choice order (Pezeshkpour & Hruschka, 2023). Further evaluations for other combinations of models can be found in Appendix A.

Table 2: Average percent edit distances between completions from `davinci-003` versus completions from the three other models and two datasets.

| Dataset | davinci-003 / ada-001 | davinci-003 / babbage-001 | davinci-003 / curie-001 | davinci-003 / Dataset |
|---|---|---|---|---|
| Wikipedia | 72.8% | 69.7% | 65.0% | 70.3% |
| DailyDialog | 75.8% | 75.1% | 74.2% | 79.0% |

## 3.2 All Model Sizes Perform Poorly At Distinguishing Their Own Completions

Figure 2 shows GPT-3's hypothetical consistency rates averaged over all few-shot prompts. Notably, all model sizes smaller than `davinci-003` perform at about random chance on this task, regardless of how many words of the original completion the model is tasked with predicting. `davinci-003` is the only model size that consistently performs above random chance, but even then its accuracy ranges only from 26% to 31% for Wikipedia and 30% to 37% for DailyDialog. We might also expect hypothetical consistency rate to increase as a function of $m$ (because longer sequences provide more information for the model to use to distinguish its own completion from other sequences), but we do not observe this trend for these models. In Appendix A, however, we do observe this trend for `gpt4`. It appears that higher-capacity and generally more powerful models are overall more hypothetically consistent.

We also inspect the frequency with which each model selects each possible answer choice, as shown in Figure 3. For a highly hypothetically consistent model that has not memorized the dataset, we would expect it to

---

[2]We select this particular set of models since it is the most recent set of text completion (rather than chat) models available for each size of GPT-3. However, we also compare against `text-davinci-001` and `gpt4` in Appendix A. We separate `davinci-001` and `gpt4` into separate analyses because `davinci-001`, `davinci-003`, and `gpt4` often output the same original completions. In our multiple-choice set-up, this results in duplicate answer choices. As such, we only ever include completions from **one** of `davinci-001`, `davinci-003`, and `gpt4` in any hypothetical consistency prompt. Since `gpt4` is a chat model whereas the rest of the models are completion models, we choose to analyze the completion models in the main text and leave comparison against `gpt4` for the Appendix.

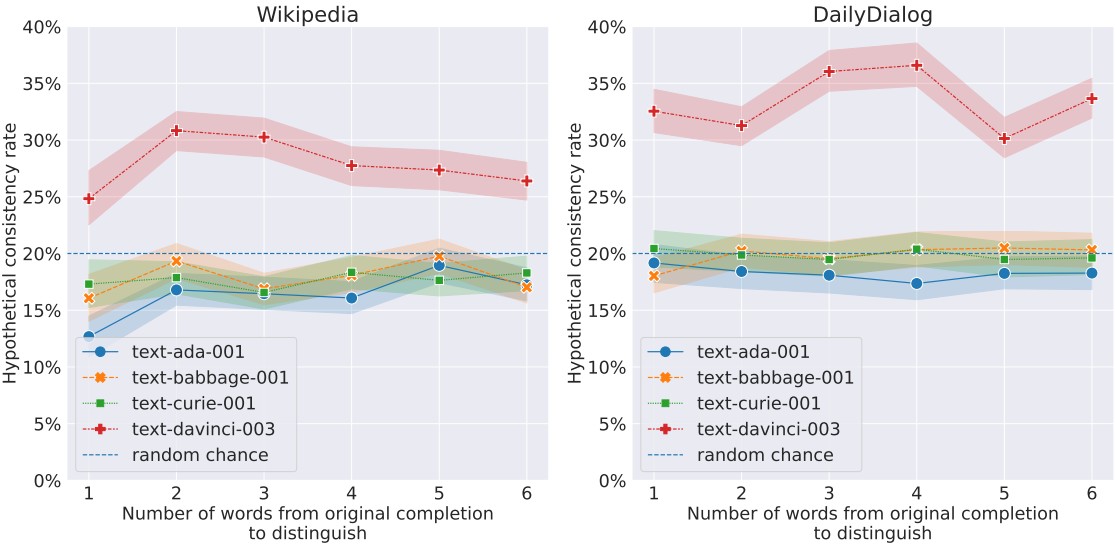

Figure 2: Hypothetical consistency rates on multiple-choice self-knowledge prompts for the Wikipedia and DailyDialog datasets, across the four GPT-3 model sizes. Each line is the average taken across all $k$-shot prompts, for $k \in [1, \cdots, 10]$. The shaded region represents the 95% confidence interval computed with nonparametric bootstrapping. The label "number of words from original completion to distinguish" corresponds to the quantity $m$ in Table 1.

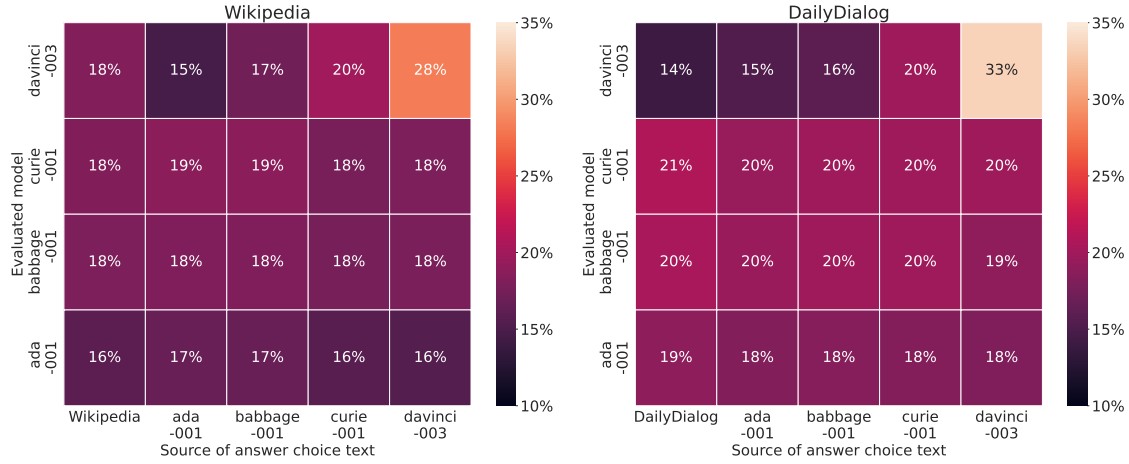

Figure 3: A more detailed breakdown of the numbers in Figure 2: the percentage of the time that each model selects each possible answer choice when prompted with a hypothetical consistency prompt, averaged across all prompts (*i.e.* across all $m$, the number of words that the model is asked to predict; and $k$, the number of few-shot examples). The columns labeled "Wikipedia" and "DailyDialog" correspond to the answer choice containing the completion from the original dataset. Model outputs that could not be parsed into an answer choice are not included.

select the answer choice corresponding to its own completion the vast majority of the time, and to only select the other answer choices a negligible proportion of the time. However, for both tasks, only `davinci-003` demonstrates a noticeable preference for its own completion over others. Furthermore, none of the other models display a preference for any other answer choice, including the completions of the other models and the continuation from the original dataset. Despite our use of multiple prompt formats, `ada-001`, `babbage-001`, and `curie-001` all make random choices and cannot predict their own completions.

`davinci-003`'s moderate hypothetical consistency also cannot easily be attributed to dataset memorization, for a couple of reasons. Firstly, we selected only hypothetical consistency prompts for which all five answer choices were distinct – ergo, `davinci-003`'s completion could not have been identical to that of either Wikipedia or DailyDialog. Secondly, we also computed the average percent edit distance (the edit distance divided by the length of the longer string) between the completions of `davinci-003` and the completions of the smaller models and datasets, as shown in Table 2. Across both datasets, the `davinci-003` completions are on average more than 70% different from the dataset completions. Furthermore, the edit distances in Table 2 do not have high variance in each row, indicating that `davinci-003`'s completions are not significantly more different from or similar to one particular source than the others.

## 4 Evaluating Compositional Self-Consistency

Next, we evaluate compositional self-consistency on the tasks of arithmetic and semantic parsing, since these are tasks for which valid compositional reasoning are important. Both tasks can be framed as computational graphs that are directed and acyclic, with each node having at most one parent – *i.e.* trees. To evaluate compositional consistency, we store the model's answer to the expression represented by each individual sub-tree and generate compositional consistency prompts by creating copies of the original prompts where a single sub-tree expression has been replaced by the model's output for that sub-tree. If a model is compositionally consistent, then it should give the same answer to the original expression as to the copy with the replaced sub-tree. Below, we further describe the experimental setup and give examples for each of the tasks.

### 4.1 Experimental Setup

We evaluate compositional self-consistency across six models (`ada-001`, `babbage-001`, `curie-001`, `davinci-001`, `davinci-003`, and `gpt4`) and two tasks. Experiments for the first five models were run between Aug. 2022 and Jun. 2023, and experiments for `gpt4` were run between May and Jun. 2023.

**Synthetic Arithmetic** We generate a set of 400 randomly-nested arithmetic expressions as the initial prompts. We then collect model completions for all possible sub-expressions of each expression using $k$-shot prompts, with $k$ ranging from 3 to 10. We randomly select the in-context examples for each prompt. The arithmetic expressions have a maximum nesting depth of 5 with a nesting probability of 0.5, and each nested expression is enclosed in parentheses to avoid ambiguity. Operators and operands are randomly selected with uniform probability from the sets $\{+, -, /, \times\}$ and $[1, 2, \cdots, 999]$, respectively.

For example, for the original arithmetic expression "$(2 \times 3) + (6/2)$," we prompt each model with the following three sub-expression prompts, using parentheses to force the correct order of operations:

$$p_1 = \texttt{"Q: } 2 \times 3 \texttt{ \textbackslash n A:"}$$
$$p_2 = \texttt{"Q: 6/2 \textbackslash n A:"}$$
$$p_3 = \texttt{"Q: } (2 \times 3) + (6/2) \texttt{ \textbackslash n A:"}$$

For each non-root sub-expression (*i.e.* $p_1$ and $p_2$), we then create a new *compositional consistency prompt* by replacing that sub-expression in the original expression (*i.e.* $p_3$) with the model's completion. For the previous example, if the model answered $p_1$ and $p_2$ correctly, this would result in the following two compositional consistency prompts:

$$p_{CC}^{(1)} = \texttt{"Q: 6 + (6/2) \textbackslash n A:"}$$
$$p_{CC}^{(2)} = \texttt{"Q: } (2 \times 3) + 3 \texttt{ \textbackslash n A:"}$$

For this example, we then compute a model's *compositional consistency rate* as the proportion of the time that the model's output for $p_i$ is correct, and its outputs for $p_{CC}^{(i)}$ and $p_3$ are the same.

**GeoQuery** GeoQuery (Zelle & Mooney, 1996) is a semantic parsing dataset consisting of 880 natural language questions about US geography, paired with FunQL (Kate et al., 2005) parses of those questions. Similar to the synthetic arithmetic task, we first collect model parses for the spans corresponding

to each sub-parse of a sample of 400 GeoQuery training examples via $k$-shot prompts, for $k$ ranging from 3 to 10. We randomly select the in-context examples for each prompt. For example, consider the GeoQuery example "Which state has the city with the most population?" with corresponding FunQL `state(loc_1(largest_one(population_1(city(all)))))`. Then two of the initial prompts we create include the following:

$p_1 =$ "Create a FunQL query for the following question: 'Which state has the city with the most population?' A: "

$p_2 =$ "Create a FunQL query for the following question: 'city with the most population' A: "

where each prompt is sourced from a non-leaf sub-parse of the original gold FunQL expression (and the other sub-parses are omitted here for brevity).

Since evaluating whether $g_{p_\theta}(f(p', p)) \sim g_{p_\theta}(f(p', g_{p_\theta}(p))$ would involve interleaving natural language with FunQL in this case, we instead measure compositional consistency as the instances where the model's parse for $p_2$ is correct (*i.e.* $p_2 \sim g_{p_\theta}(p_2)$) and is a sub-parse of its parse for $p_1$ (regardless of whether the parse for $p_1$ is correct). This second condition is a slightly more relaxed version of the condition that $g_{p_\theta}(f(p', p)) \sim g_{p_\theta}(f(p', g_{p_\theta}(p))$, where we instead assess whether $g_{p_\theta}(p)$ is being used in $g_{p_\theta}(f(p', p))$. Due to this relaxation, the rate we measure here is an upper bound on the true compositional consistency rate.

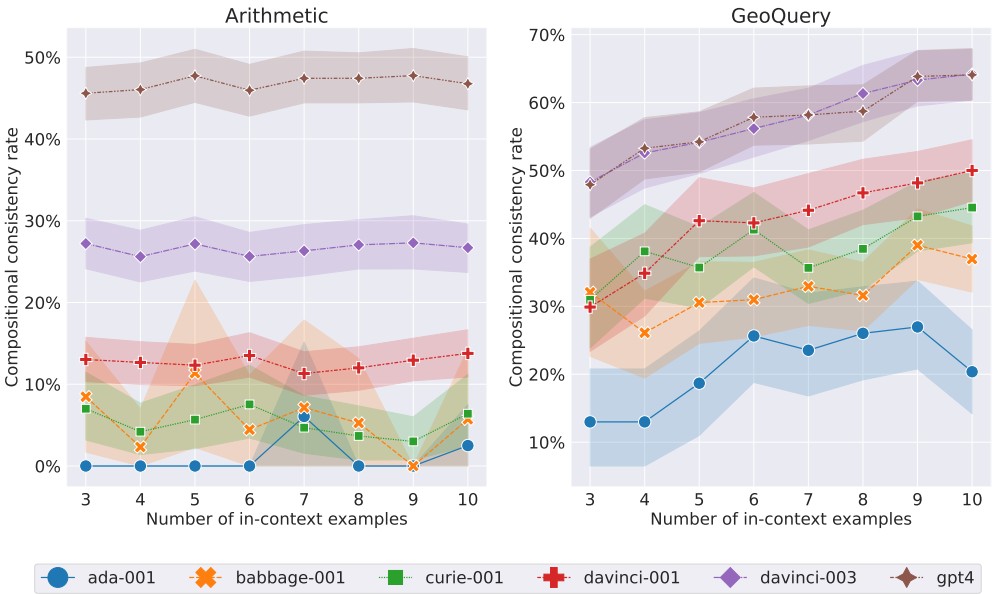

Figure 4: Compositional consistency rates versus the number of in-context examples on the arithmetic and GeoQuery tasks. The shaded region represents the 95% confidence interval computed with nonparametric bootstrapping.

## 4.2 Results

The compositional consistency rates for all six models are shown in Figure 4. While `davinci-003` and `gpt4` exhibit the highest compositional consistency rates, both are compositionally consistent less than 50% of the time on the arithmetic task and less than 65% of the time on the semantic parsing task. However, all models appear to improve in compositional consistency on the GeoQuery task as the number of in-context examples increases. Furthermore, `gpt4` exhibits significantly more compositional consistency on the arithmetic task than even `davinci-003`. Taken together, these results suggest that both increasing the model capacity and the number of in-context examples can offer some improvements in compositional consistency.

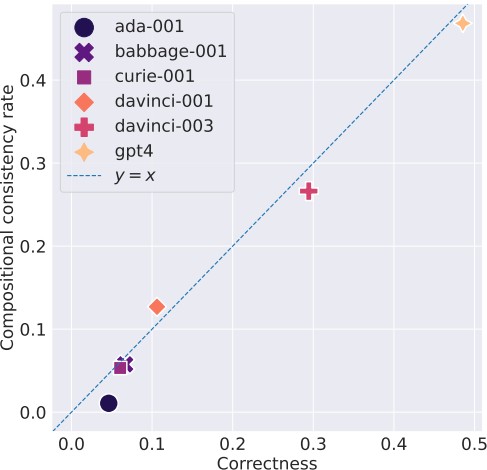

Figure 5: The correctness versus compositional consistency rate of each type of GPT-3 or GPT-4 model on the arithmetic task.

For instances where the models are compositionally inconsistent, we can analyze the sources of the inconsistency. For arithmetic, 90% of compositional inconsistencies are caused by the final answer not matching the answer of the compositional transformation, despite the answer to the sub-expression being correct (*i.e.* $p \sim g_{p_\theta}(p)$ but $g_{p_\theta}(f(p',p)) \not\sim g_{p_\theta}(f(p', g_{p_\theta}(p)))$, or case (2) in the definition of compositional inconsistency in Definition 2.7). For GeoQuery, approximately 59% of compositional inconsistencies result from the parse of the sub-tree not being included in the parse of the parent tree. For example, suppose that the parent tree is represented by the query "how many people live in Texas?" and the replaced subtree corresponds to the query "Texas." A model might correctly output the parse `stateid('texas')` for the latter but then output the parse `'population(state(name("texas")))'` for the parent tree query, which is inconsistent with the parse of the subtree. In the other approximately 41% of cases, the compositional inconsistencies on GeoQuery are caused by an incorrect parse of the child node (*i.e.* $p \not\sim g_{p_\theta}(p)$, or case (1) of compositional inconsistency in Definition 2.7).

For tasks where a precise definition of correctness does exist, such as arithmetic, it can be useful to understand the relationship between correctness and compositional consistency. Figure 5 shows this relationship for all four model sizes. There exists a notable linear relationship between correctness and compositional consistency, but all models except for `davinci-001` are slightly more correct than consistent. This indicates that models may output correct answers for the final expression but not for intermediate steps, or vice versa. Nonetheless, training LLMs to optimize solely for correctness appears to be helpful for improving compositional consistency in such tasks. For other tasks where a precise definition of correctness does not exist, this may not be as feasible a solution.

## 5   Related Work

Our work is inspired by an extensive body of literature that has defined and evaluated model consistency in a variety of ways. Elazar et al. (2021) defines consistency as the ability for the LLM to give consistent responses to semantically equivalent contexts, such as paraphrased contexts. Jang et al. (2022) supplements this definition with multiple other categories of logical consistency, such as negational, symmetric, transitive, and additive consistency. Similar to our results, they show that many modern LLMs do not exhibit strong consistency according to these definitions.

Yet other work has highlighted the inconsistency of LLM predictions across paraphrases of the same input for a variety of downstream tasks, including knowledge extraction (Elazar et al., 2021; Fierro & Søgaard, 2022; Newman et al., 2022), truthfulness (Raj et al., 2022), summarization (Kryscinski et al., 2020), and natural language understanding (Jang et al., 2021; Zhou et al., 2022). Various remedies have been proposed for this

issue – Elazar et al. (2021) proposes a novel consistency loss that minimizes the 2-sided KL divergence between paraphrases, Jang et al. (2021) proposes using multi-task training with paraphrase identification, and Newman et al. (2022) proposes training additional adapter layers to map paraphrased prompts to the same continuous representation. On the other hand, Dziri et al. (2023) suggest that the failure of LLMs to consistently reason correctly on compositional tasks is an intrinsic characteristic of the Transformer architecture – as the average parallelism of a compositional task increases, the expected error of the Transformer increases exponentially.

Our work augments the past literature by formally defining two new types of logical consistency that are crucial for valid multi-step reasoning and that have not been studied before. We additionally validate the poor performance of modern LLMs on these new types of consistency and further the understanding of why LLMs fail to generalize well on compositional tasks.

## 6    Conclusion

We have proposed two types of language model self-consistency that are important for the reliability and logically valid reasoning of LLMs on multi-step tasks. Despite the GPT-3 and GPT-4 models' generally impressive performance on a wide variety of tasks, these models still perform inconsistently on both hypothetical and compositional consistency prompts, although larger models appear to perform better. This furthers our understanding of how these otherwise impressive LLMs fail to generalize well on compositional tasks and suggests an additional reason not to trust the outputs of LLMs on complex compositional tasks, especially without extensive empirical validation. Further work is required in order to improve the logical consistency of LLM reasoning, and to investigate whether novel training techniques or further scaling improve hypothetical or compositional consistency.

## Acknowledgements

We are grateful to Eugene Choi, Richard Pang, and Nikita Nangia for helpful discussions and feedback about the design and implementation of this work. This work was supported by National Science Foundation Awards 1922658 and 2046556. CZ is supported by the DARPA PTG program. Any opinions, findings, and conclusions or recommendations expressed in this material are those of the author(s) and do not necessarily reflect the views of the National Science Foundation and DARPA. KC is additionally supported by 42dot, Hyundai Motor Company (under the project Uncertainty in Neural Sequence Modeling) and the Samsung Advanced Institute of Technology (under the project Next Generation Deep Learning: From Pattern Recognition to AI). This project has also benefited from financial support to SB by Eric and Wendy Schmidt (made by recommendation of the Schmidt Futures program), Open Philanthropy, and Apple. We also thank the NYU High-Performance Computing Center for in-kind support and OpenAI for providing access to and credits for their models via the API Academic Access Program.

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

# A Appendix

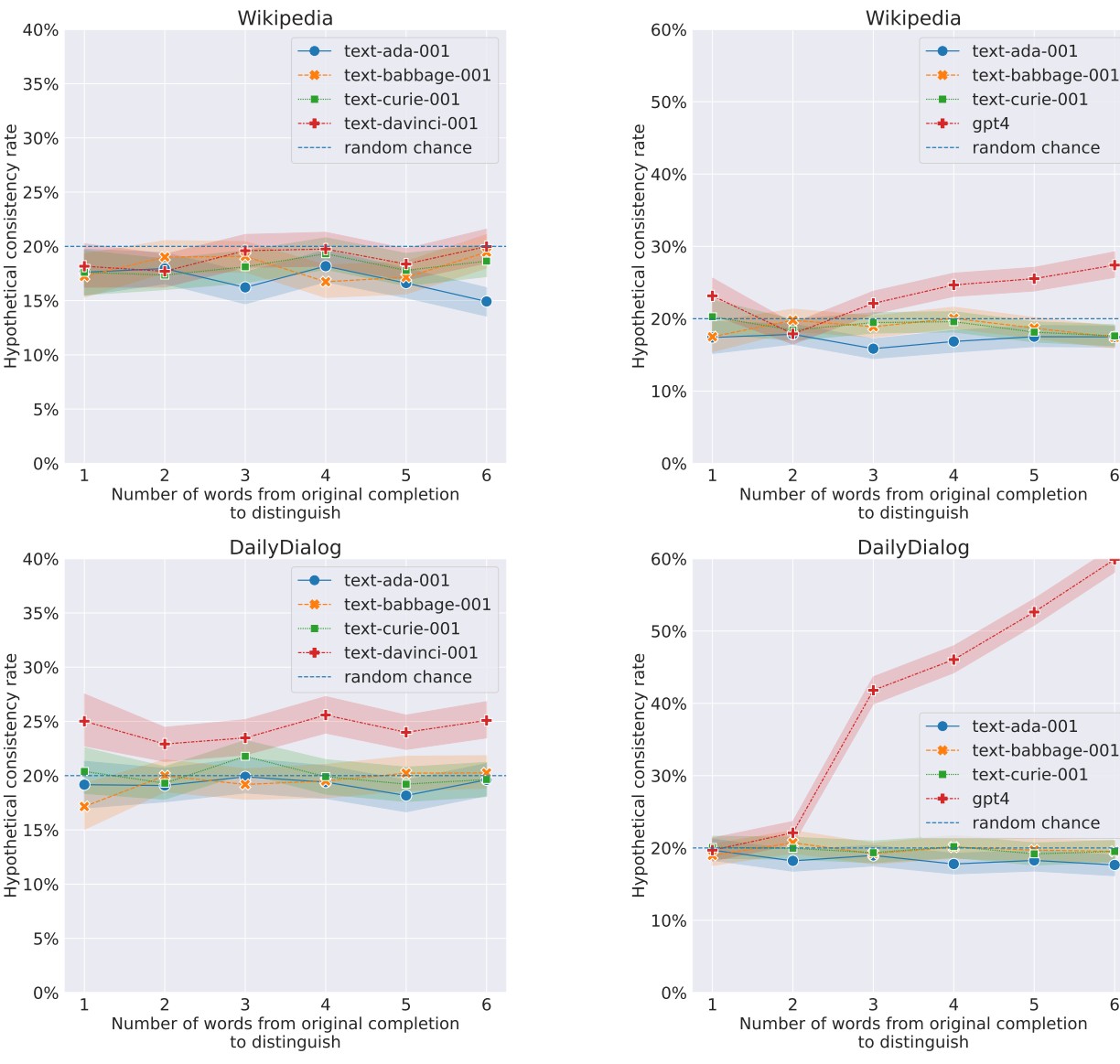

(a) Hypothetical consistency rates with answer choices generated by `ada-001`, `babbage-001`, `curie-001`, and `davinci-001`.

(b) Hypothetical consistency rates with answer choices generated by `ada-001`, `babbage-001`, `curie-001`, and `gpt4`.

Figure 6: Hypothetical consistency rates on multiple-choice hypothetical consistency prompts for the Wikipedia and DailyDialog datasets. Each multiple-choice prompt contains answer choices generated by the designated four models and an additional answer choice containing the actual continuation of the prompt in the dataset. Each line is the average taken across all $k$-shot prompts, for $k \in [1, \cdots, 10]$. The shaded region represents the 95% confidence interval computed with nonparametric bootstrapping. The label "number of words from original completion to distinguish" corresponds to the quantity $m$ in Table 1.

**Comparison of Hypothetical Consistency Against `davinci-001` and `gpt4`** We also run the same hypothetical consistency experiments on `davinci-001` and `gpt4`. Hypothetical consistency rates for `davinci-001` versus the smaller models are shown in Figure 6a, where trends are similar, but `davinci-001` performs at random chance on Wikipedia, like all the other `-001` series models. On DailyDialog, however, `davinci-001`

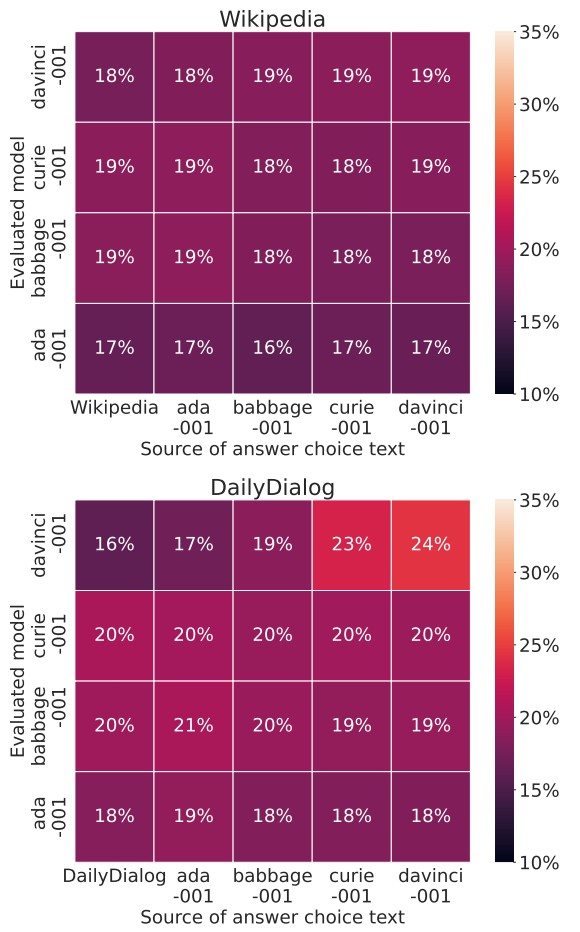

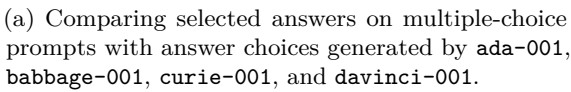

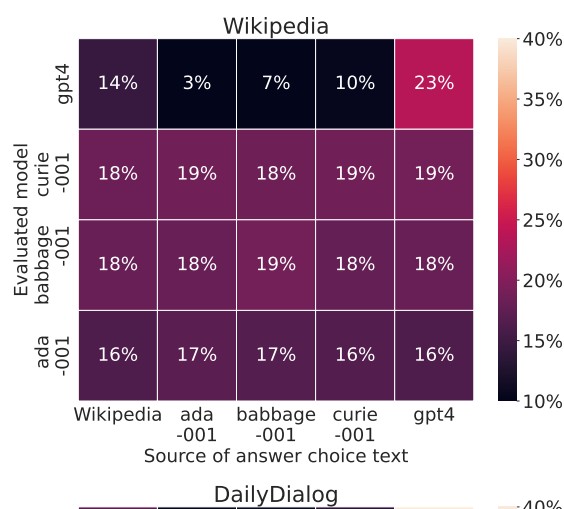

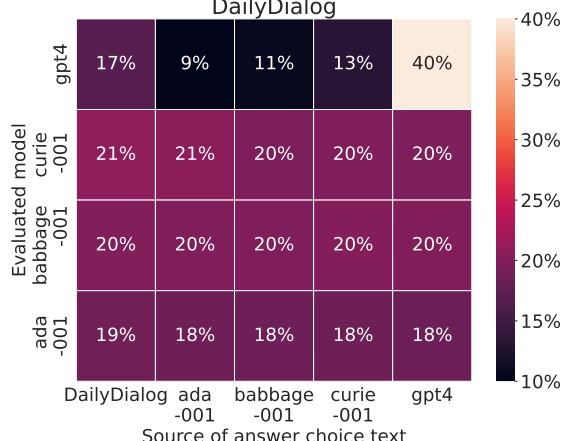

(a) Comparing selected answers on multiple-choice prompts with answer choices generated by `ada-001`, `babbage-001`, `curie-001`, and `davinci-001`.

(b) Comparing selected answers on multiple-choice prompts with answer choices generated by `ada-001`, `babbage-001`, `curie-001`, and `gpt4`.

Figure 7: The proportion of the time that each model (`ada-001`, `babbage-001`, `curie-001`, and `davinci-001`) selects each possible answer choice when prompted with a hypothetical consistency prompt. Model outputs that could not be parsed into an answer choice are not included. The columns labeled "Wikipedia" and "DailyDialog" correspond to the answer choice containing the completion from the original dataset. Model outputs that could not be parsed into an answer choice are not included.

performs noticeably better than all the other model sizes. Similar trends occur in Figure 7a, where most models are equally likely to select each answer choice on the Wikipedia dataset, and `davinci-001` is more likely to select either its own or `curie-001`'s completion on the DailyDialog dataset.

In contrast, when `gpt4` is tasked with distinguishing its own completions from those of `ada-001`, `babbage-001`, `curie-001`, and the dataset, `gpt4` performs notably better than both `davinci-001` and `davinci-003` on DailyDialog, reaching 59.9% hypothetical consistency when the number of words to distinguish is 6 (Figure 6b). However, its hypothetical consistency rate on Wikipedia is comparable to that of `davinci-003` (Figure 2), ranging from 17.9% to 27.4%. Figure 7b also demonstrates that `gpt4` is significantly more likely to select its own completion than the other models are.

It is unclear why `gpt4` is more consistent on DailyDialog than previous models of similar capacity (*i.e.* `davinci-001` and `davinci-003`). Little is known about `gpt4`'s architecture or training, aside from its multimodal abilities and training via reinforcement learning from human feedback (RLHF, OpenAI, 2023).

Since `davinci-003` was also trained with RLHF (OpenAI), it is possible that other changes in architecture or training may have also contributed to the significant improvement in hypothetical consistency.

Table 3: Average percent edit distances between completions from `davinci-001` versus completions from the three other models and two datasets.

| Dataset | davinci-001 / ada-001 | davinci-001 / babbage-001 | davinci-001 / curie-001 | davinci-001 / Dataset |
|---|---|---|---|---|
| Wikipedia | 73.6% | 71.1% | 64.4% | 71.4% |
| DailyDialog | 71.4% | 70.1% | 69.3% | 79.4% |

Table 4: Average percent edit distances between completions from `gpt4` versus completions from the three other models and two datasets.

| Dataset | gpt4 / ada-001 | gpt4 / babbage-001 | gpt4 / curie-001 | gpt4 / Dataset |
|---|---|---|---|---|
| Wikipedia | 74.1% | 71.7% | 68.9% | 68.9% |
| DailyDialog | 81.3% | 80.1% | 77.8% | 81.3% |

It is also unlikely that `gpt4`'s improvements in hypothetical consistency on DailyDialog can be attributed to dataset memorization. Firstly, we selected only prompts from both Wikipedia and DailyDialog for which all five answer choices were distinct, so `gpt4`'s completion could not have been identical to that of the original DailyDialog dataset. Secondly, we computed the average percent edit distance (the edit distance divided by the length of the longer string) between the completions of `gpt4` versus the completions of the smaller models and the datasets, which are shown in Table 4. The average percent edit distance between `gpt4` completions and DailyDialog continuations was 81.3%, indicating that `gpt4` was generating strings that were substantially different from the dataset. Similar trends were found when comparing `davinci-001` and `davinci-003` against the completions of the other models and the datasets (Tables 3 and 2).

