# OpenReview forum: "Two Failures of Self-Consistency in the Multi-Step Reasoning of LLMs"
_TMLR — Accepted by TMLR_

### Review · Reviewer_sPSQ · 2023-10-17

**Summary Of Contributions:**

The paper studies various GPT-3 and sometimes GPT-4 LLMs for consistency of answers. Two types of consistency are considered: consistency on hypothetical transformations and compositional self-consistency. The Wikipedia and DailyDialog datasets are used for the former, while the latter is evaluated on Arithmetic and GeoQuery tasks.

**Audience:**

Yes

**Broader Impact Concerns:**

No.

**Claims And Evidence:**

No

**Requested Changes:**

See above.

**Strengths And Weaknesses:**

Strengths: Studying the capabilities of popular LLMs is an important topic since LLMs are being used in an ever-increasing range of applications. Mapping out their limitations allows the research community to improve their quality and manage expectations concerning the tasks we expect such models to perform well.

Weaknesses:
* The formalism introduced in Section 2 is unclear. It only becomes clearer when the reader transitions to Sections 4 and 5. In particular:
	* The operator $\sim$ depends on an informal notion of "semantic equivalence".
	* This section has some problems with quantifiers. For instance, the second bullet on the 3rd page should have "for all $p'$" (instead of "for all $p,q\in\mathcal{V}^*$), or definitions (e.g., Definition 2.6 where from the formulation it seems that the definition of compositional consistency is conditional on $p$ and $p'$).
	* The Definition 2.4 of hypothetical consistency (HC), $g_{p_\theta}(f(p',p))\sim g_{p_\theta}(p)$, does not seem to match well with prompt templates shown on page 4 (mid-page or Table 1). For instance, it seems to generate a type mismatch, where an answer to a hypothetical prompt seems to be A/B/C/D/E, while the model's output to the raw prompt is composed of several words (e.g., "notable figures whose"). Additionally, the reader may find the best definition of HC in the last sentence of the introduction to Section 3 (page 5).
	* Definition 2.6 (consistency over compositional transformations) includes a property $p\sim g_{p_\theta}(p)$ which is confusing if one thinks about $p=$"This quilt begun in 1856 when she was seventeen includes the autographs on top of the blocks of many known celebrities and politicians of the day. Other" and $g_{p_\theta}(p)=$"notable figures whose". This makes sense only in Section 4, where e.g., the output of $g_{p_\theta}$ corresponds to a result of parsing an arithmetic expression or a FunQL reformulation of a question about US geography.
	* The naming convention of Definition 2.6 ("consistent" vs "inconsistent") is unfortunate, as consistent and inconsistent models do not cover the space of all models (the paper recognizes this fact). It seems that a better name could be used here.
	* To sum up, this section should either be revised (to form correct mathematical definitions, provide examples, and show that everything works as intended and "types match") or removed.
* Experimental sections:
	* General comments:
		* Evaluation protocol should be clearly stated and should include discussion and preferably answers to the following issues:
			* Evaluating LLMs is a known hard problem. LLMs output sequences of tokens, and there exist multiple ways of sampling from them (a.k.a. "decoding strategies"). Additionally, the models' responses are sensitive to the phrasing of prompts and in-context examples. Analyzing choices that LLMs make might require an analysis that is invariant to the particular wording of prompts. This, then, might call for an appropriate statistical methodology.
			* Some models are fine-tuned, e.g., using reinforcement learning with human feedback (RLHF). RLHF is known to cause collapse in the models' entropy.
			* It is known that GPT models are fine-tuned on the input. This means that (a) the evaluation should be performed in short closed windows and on examples (prompts) that the model has not seen before (otherwise, there could be a data leak), (b) an appropriate discussion should be provided concerning whether the used models were trained on the datasets used for evaluation (here Wikipedia, DailyDialog, arithmetic tasks, GeoQuery), (c) the results could be hard to reproduce.
		* Figures in Sections 3 and 4 mention in-context examples, but the prompt template is not provided, and not always the number of examples is known (e.g., Figure 3). Also, there is no discussion of how the choice of in-context examples can impact the answer.
	* Section 3:
		* Figure 2 could benefit from adding a comment in the caption that "Number of words from original completition to distinguish" corresponds to $m$ in Table 1. Additionally, the shade around the curves is not explained.
		* It is not clear why Figure 2 does not include GPT-4.
		* Figure 3 could be described better, e.g., that the "Wikipedia" or "DailyDialog" column corresponds to the ground truth completion from the corresponding dataset. How are the numbers related to the plots presented in Figure 2?
		* The results provided in this section are not well explained. It is not clear what additional insight is provided in the second paragraph of Section 3.2. Some questions that could be answered are: What results should be expected before the experiment, and what are the reasons for the actual results? What shapes of the curves do we expect and why? How is it related to the greedy decoding scheme chosen in the paper (e.g., it might not generate the most likely response, and hence, the answers of other models could have a higher likelihood)? What is the uncertainty about these results (see the "Evaluation protocol" item above)?
	* Section 4:
		* Figure 4 includes multiple in-context examples.
		* The GoeQuery causes some quirks where natural language could be mixed with FunQL. This results in a relaxed consistency notion. How could this approximation impact the results of the experiments?
		* The phenomenon described in Section 4.2 concerning the arithmetic dataset should be compared to similar results for the chain of thought techniques, where the final answer does not necessarily match the provided reasoning.
		* The situation described for GeoQuery data, where the same query can be parsed in multiple ways, could call for more careful curation of queries or raise a concern about whether this is a correct dataset for the current study.

---

### Review · Reviewer_QKsC · 2023-10-25

**Summary Of Contributions:**

This paper studies different GPT models in terms of two consistency metrics: hypothetical consistency and compositional consistency. The paper formalizes the two consistencies metrics and performs detailed experiments to evaluate both of them.

Hypothetical consistency is evaluated on auto-completion tasks using recent Wikipedia text (i.e., text not used to train the GPT models). Results show that all models evaluated performed poorly in terms of hypothetical consistency.

Compositional consistency is evaluated on arithmetic and GeoQuery tasks, with similar conclusions, that the models tend perform poorly, but with a significant improvement for larger models (davinci-003 and gpt4).

**Audience:**

Yes

**Claims And Evidence:**

No

**Requested Changes:**

I would appreciate a discussion about the lack of user studies to evaluate the models, why they weren't performed and why we are able to approximate their results with the current set of experiments. In particular, this discussion should explain why the string-matching approach used in the Section 3 doesn't provide a pessimistic view of the results.

The contributions need to be toned down. Instead of claiming "why", the paper should list only "how" mistakes are made as a contribution.

I would also like to see the results from the Appendix in the main text.

**Strengths And Weaknesses:**

Strengths:

The paper does a good job formalizing the two new consistency metrics and evaluating them on a good number of models and tasks. The experiments performed in this paper are non-trivial and I appreciate the authors' effort in putting this together. Overall, the paper makes a good contribution as it shows yet another data point demonstrating logical inconsistencies of LLMs.

Weaknesses:

The paper pays a somewhat high "notation cost" for formalizing the two metrics. While reading the paper I couldn't help but wonder if there was a more friendly way of formalizing and presenting the results. The paper is heavy on notation and I had to go back and forth the paragraphs to remind myself what each of the letters meant. I even wondered whether the paper would stand by itself if it had Section 2 remove. Section 2 is very dense and it doesn't offer any new insights and I am not sure it helps the reader understand the later results of the paper.

Given the current notation used in Section 2, I would write $p^*_{\theta}$ instead of $p^*$, as it would help distinguish that we are talking about a model (even if it is the "perfect" model).

The first part of the paper (Sections 1 and 2) talks about semantically similar outputs, but Section 3 evaluates the models on exact string matching. I find that to be a serious weakness of the paper. For example, there is a chance that what is currently considered a wrong choice of the model is in fact a choice that is semantically equivalent to the model's previous choice. Perhaps the numbers shown in Section 3 are pessimistic because it only counts an answer as correct if the string is exactly the same as the model's previous output (case insensitive comparison).

I understand why the authors decided to perform such as evaluation, as it would be hard to automatically evaluate semantic equivalence. The gold standard in this case should be to run a user study to evaluate the models.

In Section 4, in addition to showing the average results of the models in terms of compositional consistency, the paper also discusses how the models make mistakes. Later, in the conclusion section, it reads as a contribution of the work: "further the understanding of why LLMs fail to generalize well on compositional tasks." I think this isn't correct. What the paper shows is *how* the LLMs make mistakes and it is far from explaining *why*.

This is an empirical paper, so I would prefer to see the results of the Appendix in the main text. If the authors were looking for a shorter paper, I would easily exchange parts of the formalization with more empirical results.

---

> ### Author Response · Authors · 2023-11-14
> **Thank you for the feedback! We have included revisions in the draft (in red font) and provided responses below.**
>
> Thank you for carefully reading and providing feedback to our paper -- we greatly appreciate it. We provide our responses below:
> - "The paper pays a somewhat high "notation cost" for formalizing the two metrics…Section 2 is very dense and it doesn't offer any new insights and I am not sure it helps the reader understand the later results of the paper."
>     - We agree that formalized definitions can sometimes be dense – however, our precise notation also removes any ambiguities that may exist in the informal definitions (i.e. the ones given in Section 1). This notation also allows us to give concise demonstrations of how the experiments correspond to the definitions (e.g. the example at the bottom of page 4), proofs of how one type of consistency may be related to another (Claim 2.6), precise definitions of each type of consistency rate (e.g. first paragraph of Section 3), and pointers to the exact ways in which the definitions may need to be relaxed for practical evaluation (e.g. last paragraph of Section 4.1). Without the notation, these explanations would be rather verbose and possibly ambiguous.
> - "The first part of the paper (Sections 1 and 2) talks about semantically similar outputs, but Section 3 evaluates the models on exact string matching…Perhaps the numbers shown in Section 3 are pessimistic because it only counts an answer as correct if the string is exactly the same as the model's previous output (case insensitive comparison)."
>     - For the exact matching in the hypothetical consistency evaluations, we are matching on the **letter** (e.g. "E") corresponding to the correct answer choice, rather than the model's complete previous output string (e.g. "notable figures whose"). Given that there are in-context examples to glean the format from, this is a considerably easier task than generating the exact same output string as with the original prompt.
>     - However, thank you for the note about the pessimistic estimate! It is likely that our numbers may still be an under-estimate of the true hypothetical consistency rates, since there may exist other outputs that are semantically equivalent but lexically distinct from the model's original output. Nevertheless, we find it more beneficial to analyze the lower bound rather than the upper bound, as this yields more insights about the failure modes of the LLM, rather than only the success cases. Analyses of these failure modes can inform future model innovations.
>     - We have also added a paragraph at the top of page 5 to discuss the possibility that our estimates are lower bounds on the true hypothetical consistency.
> - "What the paper shows is how the LLMs make mistakes and it is far from explaining why…Instead of claiming "why", the paper should list only "how" mistakes are made as a contribution."
>     - Thank you for the note, we have changed this wording in our paper.
> - "The gold standard in this case should be to run a user study to evaluate the models."
>     - We agree that running a user study to evaluate the semantic equivalence of the outputs in the hypothetical consistency experiments would likely provide the best signal. Furthermore, the noisy nature of human annotations would require multiple annotations per pair of model outputs, resulting in a large number of total annotations. We sadly do not currently have the resources for this type of user study, but we agree that future studies should attempt to develop a more robust method of assessing semantic equivalence.
>     - Nevertheless, this is mitigated in multiple ways. (1) In our hypothetical consistency experiments, the LLM is tasked only with outputting the letter of the correct answer choice, leaving no ambiguity in whether it has selected the choice that is the *most* semantically equivalent to its original output. (2) In our compositional consistency evaluations, we have designed our experiments around tasks that tend to have precise answers and less room for lexical variation (e.g. numbers or FunQL syntax).
> - "This is an empirical paper, so I would prefer to see the results of the Appendix in the main text. If the authors were looking for a shorter paper, I would easily exchange parts of the formalization with more empirical results."
>     - We understand that it is sometimes easier to view all the results when they are all in the main text rather than in the Appendix. In this case, we have referenced the supplemental results in the main text (*e.g.* footnote 2 on page 5) and kept the plots in the Appendix to prevent disrupting the flow of the paper. All main results that we discuss extensively are located in the main text.

---

### Review · Reviewer_1XQ6 · 2023-10-31

**Summary Of Contributions:**

The paper builds on the recent work on self-consistency of LLMs and proposes two new kinds of consistency checks - hypothetical transformation and compositional transformations. Hypothetical transformations include prompts such as those asking for completion of the prompt and response to check whether LLM produces same or similar response. Compositional transformations replace the intermediate steps with the response of the LLM to check whether the LLM response remains conformant for these changes.

**Audience:**

No

**Broader Impact Concerns:**

There are no broad impact concern. The introduction section brings out broader impact well.

**Claims And Evidence:**

No

**Requested Changes:**

Please see the weaknesses pointed out in the section above.

**Strengths And Weaknesses:**

+ The use of semantic consistency to evaluate LLM's trustworthiness has gained a lot of attention and the paper makes interesting observations on how two specific transformations can be useful in this context.

- Formalization is rather trivial and the composition formalization is illustrated via an example. The reviewer understands the challenge of formalizing arbitrary composition transformation since more involved prompts could need higher order logic specification. The notations make a simple idea appear unnecessarily complicated without adding mathematical rigor.

- The use of "multiple choice" pattern makes the application of this approach difficult because generating reasonable such choices automatically will be difficult. Using completions of other models makes the test a bit biased. We are no longer checking consistency of a given model but checking consistency modulo response of other models. Correlated failures (which are likely given similar data) can make this problematic.

- The use of arithmetic and geoquery datasets for compositional transformation appears to highlight the limited utility of this approach when compositions are easy to identify.

- Overall, the paper is light on novelty. Two consistency checking transformations are used to evaluate a set of models on a set of datasets. Conclusions appear rather straightforward and unsurprising - "all models fail at distinguishing their own completions". The claims and corresponding evidences in the paper need to be strengthened.

---

> ### Author Response · Authors · 2023-11-14
>
> We thank the reviewer for volunteering their time and energy to offer their feedback on our work. Our responses to the concerns and requested changes are below:
>
> - "The notations make a simple idea appear unnecessarily complicated without adding mathematical rigor."
>     - We understand that formalized definitions can sometimes appear complicated – however, our precise notation also removes any ambiguities that may exist in the informal definitions (i.e. the ones given in Section 1). This notation also allows us to give concise demonstrations of how the experiments correspond to the definitions (e.g. the example at the bottom of page 4), proofs of how one type of consistency may be related to another (Claim 2.6), precise definitions of each type of consistency rate (e.g. first paragraph of Section 3), and pointers to the exact ways in which the definitions may need to be relaxed for practical evaluation (e.g. last paragraph of Section 4.1). Without the notation, these explanations would be rather verbose and possibly ambiguous.
> - "Using completions of other models makes the test a bit biased. We are no longer checking consistency of a given model but checking consistency modulo response of other models. Correlated failures (which are likely given similar data) can make this problematic."
>     - Our method checks hypothetical consistency with respect to both the responses of other models and continuations from the original dataset. It is unfortunately computationally infeasible to check against every possible response to the original prompt – however, we have chosen a set of options that we believe are realistic distractors. For instance, similar models may have similar biases that result in the models incorrectly selecting each others' responses to the original prompt. In addition, the answer choice corresponding to the continuation from the original dataset may have higher likelihood if it appeared in the training dataset, but in our experimental set-up it will never be the same as $g_{p_\theta}(p)$.
> - "The use of arithmetic and geoquery datasets for compositional transformation appears to highlight the limited utility of this approach when compositions are easy to identify."
>     - We agree that compositions are easier to identify in these tasks, and it would be useful for future work to explore compositional consistency in tasks with more complex or open-ended compositions (e.g. long-term planning). However, our results here indicate that LLMs already struggle with consistency in these easier compositional tasks. This gives a helpful upper bound on the compositional consistency of more complex and open-ended tasks, even if we do not yet have the adequate experimental set-up for evaluating them. We leave the evaluation of compositional consistency in more open-ended tasks to future work.
> - "Overall, the paper is light on novelty…Conclusions appear rather straightforward and unsurprising."
>     - Indeed, we do not claim these results are surprising in any way -- we also believe that a scientific contribution need not be surprising nor non-straightforward to be meaningful. Without careful experimental design and evaluation, our claims would be merely conjecture. Even if these claims are intuitive and not particularly surprising, it is still impactful to both formalize the claims and empirically validate them. The kinds of consistency we investigate are important for the correct logical reasoning of LLMs, and have not -- to the best of our knowledge -- been studied before. Improving upon and evaluating these kinds of consistencies will aid in the development of LLMs with more robust reasoning abilities.
>     - Furthermore, these results may still be surprising in the sense that LLMs have been broadly lauded for their fluent outputs and impressive capabilities across a variety of tasks. Yet our results show that LLM reasoning still often lacks logical soundness even on some very basic tasks like arithmetic.

---

> > ### Comment · Reviewer_1XQ6 · 2023-12-05
> > **Thanks**
> >
> > Thank you for your clarifications.

---

### Review · Reviewer_3Fto · 2023-12-29

**Summary Of Contributions:**

The paper explores the importance of self-consistency in large language models (LLMs) for multi-step reasoning tasks. It introduces two types of self-consistency, hypothetical consistency and compositional consistency, and demonstrates that various versions of GPT-3/-4 models exhibit poor consistency rates in both types. The paper formalizes the definitions of consistency and prompt transformations and presents empirical evidence of low consistency rates in LLMs.

**Audience:**

No

**Broader Impact Concerns:**

There are no broad impact concern.

**Claims And Evidence:**

No

**Requested Changes:**

Please see the weaknesses pointed out in the section above.

**Strengths And Weaknesses:**

Strengths:

1. Nice presentation of the figures and experimental results.

2. The paper is easy to follow.

Weaknesses:

1. From my point of view, experiments can be more sufficient. For example, is it possible to add more LLMs, such as Llama and ChatGLM, as the backbone model, for evaluation?

2. Although the paper mainly focuses on consistency, it would be better to provide an accuracy evaluation.

3. The related work part can be more complete with the latest achievements on LLMs.

---

### Decision · Action_Editor_pgc6 · 2023-12-26

**Recommendation:** Accept with minor revision

**Comment:**

The empirical evaluation of consistency is good, but the reviewers remain concerned about the complexity of the formalism introduced in Section 2 to define consistency.  Nevertheless, it is good to define consistency and inconsistency formally.  Additional comments by the reviewers include:

1. The paper continues to use consistent/inconsistent notion that does not split the space of models into two parts (i.e., there are models that are neither consistent nor inconsistent).
2. The example for Definition 2.5 provided at the bottom on page 4 assumes that the equivalence relation depends on p' (otherwise, there is no way to know what "E" corresponds to). This should be added to the list of simplifications made for the purpose of experiments that is discussed on page 5.
3. In Figure 3, it is still unclear what " averaged across all prompts" refers to (over all m and k?)
4. The confidence intervals are not described (is it bootstrap, parametric, or over how many outcomes/seeds)?
5. Some models considered in the paper are tuned on human data (IFT or PPO). This could explain why the models output the same original completions. These kinds of behaviors were alluded to in the question concerning mode collapse by RLHF.
6. In Section 3.2, it seems that gpt4 displays some interesting properties that other models do not (visualized on Figure 6b). It is strange why this was not put in the main body of the paper (this circles back to the question concerning reporting results for gpt4).
7. Since there is only one section in Appendix A, there is no need to introduce "A.1".

**Audience:**

Yes, this work will be of interest to anyone working on advancing and evaluating LLMs.

**Claims And Evidence:**

The paper studies two forms of consistency in LLMs and claims that state of the art LLMs generally fail to be consistent.  This claim is supported by suitable experimental evaluation that shows a low degree of consistency for a variety of tasks and LLMs.  The paper also contributes a formalization of consistency in terms of template prompts, composition, semantic equivalence and greedy decoding.